# Identification of Chromoblastomycosis and Phaeohyphomycosis Agents through ITS-RFLP

**DOI:** 10.3390/jof10020159

**Published:** 2024-02-18

**Authors:** Gabriel S. M. Sousa, Rodrigo S. De Oliveira, Alex B. De Souza, Ruan C. Monteiro, Elaine P. T. E. Santo, Luciano C. Franco Filho, Silvia H. M. Da Silva

**Affiliations:** 1Programa de Pós-Graduação em Biologia de Agentes Infecciosos e Parasitários, Instituto de Ciências Biológicas, Universidade Federal do Pará, Belém 66075-750, Brazil; 2Laboratório de Micoses Superficiais e Sistêmicas, Seção de Bacteriologia e Micologia, Instituto Evandro Chagas, Ananindeua 67030-000, Brazil; rodrigooliveira@iec.gov.br (R.S.D.O.); alexbritosouza@yahoo.com.br (A.B.D.S.); elainetavares@iec.gov.br (E.P.T.E.S.); lucianofranco6@gmail.com (L.C.F.F.); silviasilva@iec.gov.br (S.H.M.D.S.); 3Laboratory of Emerging Fungal Pathogens, Universidade Federal de São Paulo, São Paulo 04023-062, Brazil; rcmonteiro@unifesp.br

**Keywords:** chormoblastomycosis, phaeohyphomycosis, PCR-RFLP, *Herpotrichiellaceae*, sequencing, phenotypic, molecular

## Abstract

Chromoblastomycosis (CBM) and phaeohyphomycosis (FEO) are infections caused by melanized filamentous fungal agents, primarily found in tropical and subtropical regions. Both infections pose significant challenges for the correct identification of the causative agent due to their morphological similarity, making conventional methods of morphological analysis highly subjective. Therefore, molecular techniques are necessary for the precise determination of these species. In this regard, this study aimed to contribute to a new methodology based on PCR-RFLP for the identification of agents causing CBM and FEO. Sequences from the *Internal Transcribed Spacer* (ITS) region were used to identify potential restriction enzyme sites in silico, followed by in vitro validation using the selected restriction enzymes. The obtained results were compared with species identification through morphological analyses and sequencing. The results demonstrated that the PCR-RFLP applied in this study accurately identified two major agents of chromoblastomycosis, *Fonsecaea pedrosoi* and *Fonsecaea monophora*, as well as *Cladophialophora bantiana* and *Exophiala dermatitidis*, both causative agents of phaeohyphomycosis. In this context, the proposed assay can complement current methods for identifying these species, aiding in diagnosis, and contributing to the proper management of these infections.

## 1. Introduction

Chromoblastomycosis (CBM) and phaeohyphomycosis (PHM) are fungal infections primarily instigated by dematiaceous fungi, predominantly classified under the *Herpotrichiellaceae* family [1,2]. CBM is categorized by the World Health Organization as a neglected tropical disease, primarily due to its higher prevalence in economically disadvantaged or developing countries situated within tropical or subtropical regions [3,4,5].

CBM prevails endemically in countries spanning Latin America, Central America, Africa, and Asia, with a notable predilection for rural laborers [2,3,6,7]. In contrast, PHM is reported in diverse nations globally and exhibits a similar affinity for tropical and subtropical climates, predominantly affecting immunocompromised individuals, thus denoting it as an opportunistic infection [1,8,9].

Chromoblastomycosis and phaeohyphomycosis are both mycoses primarily transmitted through direct contact with contaminated materials, often resulting from traumatic injuries. These materials may encompass plant thorns, branches, soil, and various other organic components [1,2].

The clinical manifestations of both infections are intricate, often mirroring other maladies, thereby complicating the process of differential diagnosis. Consequently, laboratory support is frequently imperative [8,10,11]. In this context, the prevailing diagnostic gold standard for both infections are the identification of muriform cells in the case of chromoblastomycosis and dematiaceous hyphae in the case of phaeohyphomycosis through direct mycological examination [1,2,12].

In the presumptive species-level identification, morphological analyses are conducted based on the macrocolony’s appearance and the arrangement and structure of conidial and hyphal elements [2,13,14]. However, due to the phenotypic similarities between the species responsible for CBM and PHM, such analyses become notably subjective [8,14].

Owing to this subjectivity, the genetic sequencing of the Internal Transcribed Spacer (ITS) region has emerged as the most widely employed molecular technique for identifying the agents of chromoblastomycosis and phaeohyphomycosis [1,2]. Nevertheless, this methodology is associated with high costs and is relatively inaccessible in regions where these infections are more endemic. Consequently, there is a pressing need for new, cost-effective molecular approaches suitable for resource-limited laboratories, enabling rapid and specific species identification [15].

In light of these considerations, the present study aimed to develop a molecular approach based on the restriction fragment length polymorphism (RFLP) of the *Internal Transcribed Spacer* (ITS) region for the identification of species among the primary causative agents of chromoblastomycosis and phaeohyphomycosis in the northern region of Brazil.

## 2. Materials and Methods

The experiment was divided into stages involving in silico analyses and subsequent in vitro validation, following the workflow depicted in Figure 1.

### 2.1. Clinical Strains

The isolates from chromoblastomycosis (N = 19) and phaeohyphomycosis (N = 2) examined in the present study are maintained in the mycoteca of superficial and systemic mycoses at the Instituto Evandro Chagas (IEC) in the state of Pará, Brazil. The representative fungal agents were originally obtained from clinical samples sourced from healthcare units and hospitals within the state of Pará, as evidenced in Table 1. These agents have already been molecularly identified, and their sequences have been deposited in the GenBank platform of NCBI.

### 2.2. Morphological Identification

The morphological characteristics of the isolates were assessed through microcultivation on slides, employing lactrimel agar medium, and incubated at 37 °C for 14 days. Subsequently, the slides were scrutinized with lactophenol cotton blue staining, utilizing 40× objectives, for the documentation of conidial arrangement organization [14,16].

### 2.3. DNA Extraction

The cultures were subcultured in a tube containing YPD Agar and incubated at 30 °C for 14 days for DNA extraction. The extraction process involved collecting approximately 400 mg of fungal mass, which was then added to a 2 mL tube containing a 450 μL solution composed of 150 μL lysis buffer (SDS), 150 μL homogenization buffer, and 150 μL TE buffer. Glass beads were added, and the microtube was vortexed for 30 min. Subsequently, 15 μL of proteinase K were added, and the microtube was incubated in a water bath at 57 °C for one hour. Following this, 200 μL of 5 mol/L sodium chloride was added, and the microtube was incubated again at 67 °C for 10 min.

After incubation, 600 μL of the liquid was transferred to a new microtube and purified using the phenol–chloroform–isoamyl alcohol protocol described by Campos in 2017 [17]. To achieve enhanced DNA purity, the Bioflux DNA purification kit (Hangzhou Bioer Technology Co., Ltd., Hangzhou, China) was employed in conjunction with the product obtained through the phenol–chloroform methodology, adhering to the procedures delineated in the manufacturer’s protocol.

The extracted DNA was quantified using the NanoDrop 2000© spectrophotometer (Thermo Fisher Scientific Inc.^®^, Waltham, MA, USA). We used the standard value of 1 OD = 50 µg/mL to determine the double-stranded DNA concentration. Only samples with an OD260/280 ratio between 1.7 and 2.0 were included in the study.

### 2.4. Molecular Identification through Sequencing

Molecular sequencing of the isolates was performed to confirm the previously identified species, focusing on the *ITS* region, which includes *ITS1*, *5.8S*, and *ITS2*, using *ITS1*(F) TCCGTAGGTGAACCTGCGG and ITS4(R) TCCTCCGCTTATTGATATGC primers. The PCR conditions consisted of an initial denaturation at 95 °C for 5 min, followed by 35 cycles of denaturation at 94 °C for 1 min, primer annealing at 55.5 °C for 2 min, and extension at 72 °C for 2 min. Finally, a 10 min extension phase at 72 °C was conducted [18].

The amplification reaction was carried out with 4 mM MgCl_2_, 0.4 mM of each dNTP (deoxynucleotide triphosphate), 1 mM of the primers, 0.1 μL of Taq DNA polymerase (Thermo Fisher Scientific Inc.^®^, Waltham, MA, USA), 2.5 μL of 10 mM/L BSA, and 2 μL of DNA, in a final volume of 25 μL. PCR was performed using a PX2 Thermo Hybaid thermocycler (Artisan Technology Group, Champaign, IL, USA). The amplification product was visualized by electrophoresis on a 1.5% agarose gel.

Following electrophoresis, amplicon purification was carried out using the ExoSAP-IT™ PCR Product Cleanup Reagent (Thermo Fisher Scientific Inc.^®^, Waltham, MA, USA), following the manufacturer’s recommendations. Sequencing was performed using the BigDye™ Terminator v3.1 Cycle Sequencing Kit (Thermo Fisher Scientific Inc.^®^, Waltham, MA, USA), following the manufacturer’s instructions, and the samples were sequenced on an ABI 3130/3130XL Automatic Genetic Analyzer using the dideoxnucleotide chain termination method (Thermo Fisher Scientific Inc.^®^, Waltham, MA, USA).

Identification was accomplished by aligning the sequences with those deposited in the GenBank and ISHAM databases, referencing type strains and considering a similarity value greater than 99% for species determination.

### 2.5. In Silico ITS-RFLP

In order to identify enzymes with the potential for species differentiation among the etiological agents of chromoblastomycosis and phaeohyphomycosis, in silico screening was conducted using sequences from the *ITS* region of the study’s target species (N = 60) available in the GenBank database (Appendix A). These sequences were downloaded and compiled into a MultiFasta file and aligned with the online software Mafft 7 “https://mafft.cbrc.jp/alignment/server/index.html (accessed on 20 August 2023)” [19].

Subsequently, the presence of restriction sites for commercially available enzymes was analyzed using the CLC Sequence Viewer software 8.0^®^ [20], with the goal of generating fragments of distinct sizes suitable for species differentiation while ensuring the consistency of restriction sites within the same species. Following the screening and identification of potential enzymes, the online tool NEBcutter v.3^®^ from New England Biolabs “https://nc3.neb.com/NEBcutter/ (accessed on 25 August 2023)” was employed for visualizing the fragmentation pattern in a virtual agarose gel [21].

### 2.6. ITS-RFLP

The *ITS* fragments were amplified using the primer pair ITS1F (5′-TCCGTAGGTGAACCTGCGG-3′) and ITS4R 5′TCCTCCGCTTATTGATATGC-3′), as previously described. The negative control for the reaction contained all PCR components, except the DNA, of the agents responsible for chromoblastomycosis and phaeohyphomycosis. PCR products were visualized through electrophoresis on a 1.5% agarose gel using a UV transilluminator. Subsequently, restriction enzyme digestion was conducted in a final volume of 40 μL, following the reaction conditions recommended by the manufacturer (Thermo Fisher Scientific Inc.^®^, Waltham, MA, USA).

In the single enzymatic digestion, a mixture of 3 μL of Red Buffer (10×) and 1 μL of HhaI enzyme (10 U/μL) was added for every 20 μL of PCR product. In the double digestion reaction, 1 μL of HaeIII enzyme (10 U/μL), 1 μL of HhaI enzyme (10 U/μL), and 3 μL of Red Buffer (10×) were added for every 20 μL of PCR product. A negative control reaction was also performed for each digestion, containing all reagents except the PCR product.

Both reactions were incubated at 37 °C for 1 h. The digestion products were then subjected to electrophoresis on a 3% (*w*/*v*) agarose gel for 120 min, and the fragments were visualized under a UV transilluminator.

## 3. Results

### 3.1. Morphological Identification and Sequencing of the Isolates

In this study, a total of 21 isolates originating from chromoblastomycosis and phaeohyphomycosis were utilized. The morphologic identification through the micro cultivation analysis allowed for the determination of their genus-level classification, resulting in 19 isolates belonging to the genus *Fonsecaea* sp., 1 to the genus *Exophiala* sp., and 1 to the genus *Cladophialophora* sp.

The molecular identification through the sequencing of the *ITS* region revealed that out of the 21 isolates, 15 were identified as *Fonsecaea pedrosoi*, 4 as *Fonsecaea monophora*, 1 as *Exophiala dermatitidis*, and 1 as *Cladophialophora bantiana*.

### 3.2. In Silico Analyses

Through the in silico analyses, we have concluded that the enzymes HhaI and HaeIII are capable of distinguishing between the species in this study through the restriction of the *ITS* region. These enzymes can be used in two different assays, the first being a single digestion with HhaI and the second being a double digestion with HhaI and HaeIII. The two proposed assays differ in fragmentation patterns and their ability to distinguish between the species, as shown in Figure 2.

### 3.3. ITS-RFLP with HhaI

The in silico results of the single digestion with the HhaI enzyme were compared with the fragmentation patterns obtained in vitro (Figure 3), where the results were confirmed, allowing for the differentiation between the species *Fonsecaea pedrosoi* (352, 219 bp) and *Fonsecaea monophora* (330, 188, 53 bp). The fragmentation patterns also differed for the species *Exophiala dermatitidis* (193, 193, 135, 98 bp). However, it was not possible to distinguish between the species *Fonsecaea monophora* and *Cladophialophora bantiana* in the single digestion.

### 3.4. ITS-RFLP with HhaI and HaeIII

A comparison was made between the double digestion fragmentation patterns obtained in silico and in vitro (Figure 4), where similar results to that obtained with the single digestion was achieved for all species, but with a greater power of distinction between the species *Fonsecaea monophora* (203, 130, 58, 53, 52, 48, 27 bp) and *Cladophialophora bantiana* (182, 127, 95, 59, 49, 42, 17 bp).

### 3.5. Comparative Analysis of Identification Methods

The results obtained in both enzyme restriction assays were compared to other identification methods employed in this study (morphologic and sequencing). It was observed that the ITS-RFLP method exhibited a higher species identification capability compared to the phenotypic method. Furthermore, the ITS-RFLP method demonstrated a species identification capacity similar to sequencing, as described in Table 2.

## 4. Discussion

Infections caused by melanized fungi can significantly impact people’s quality of life, leading to discomfort, pain, and, in severe cases, disfigurement. Both chromoblastomycosis (CBM) and phaeohyphomycosis (FEO), when affecting the skin and subcutaneous tissues, can be persistent and challenging to treat, often requiring prolonged therapies and, in some cases, surgical interventions [1,8,22,23,24]. In immunocompromised patients, such as those with HIV/AIDS, uncontrolled diabetes, or transplant recipients, agents causing phaeohyphomycosis can disseminate to internal organs and the central nervous system, posing a severe life-threatening risk [1,25,26].

Diagnosing both infections is notoriously challenging and difficult, with chromoblastomycosis typically taking an average of nine years to be correctly identified. In many cases, the causative species remains unidentified [3], a crucial concern since recent studies have highlighted variations in pathogenicity, virulence, tropism, and susceptibility to antifungals among these species [1,2,27]. Therefore, identifying the causative agent is essential for a more accurate prognosis [28].

The lack of identification of these agents in many cases occurs due to the phenotypic and morphological similarity of the species associated with these infections, rendering conventional morphological analysis methods nonspecific. For that reason, applying sequencing and other molecular biology techniques is necessary in most cases [1,8,14,28].

It is evident that the identification of species causing CBM and FEO poses a challenge to overcome, and the use of molecular techniques can help overcome this barrier. In this context, RFLP emerges as a simple and promising technique for the identification of CBM and FEO agents. Recent studies have shown that this technique serves as a valuable tool for identifying a variety of pathogenic fungal species, such as *Cryptococcus* sp., Dermatophytes, *Paracoccidioides* sp., and *Candida* sp. [20,29,30,31].

Currently, there is no existing literature on assays for the identification of CBM and FEO agents using RFLP, with only isolated studies of specific primers which either do not encompass all currently identified agents or lack sensitivity and specificity [32].

Due to this gap, the methodology proposed in this study aimed to contribute to the standardization of a species-specific diagnostic method using RFLP of the ribosomal DNA *ITS* region, known for its high conservation in fungi and currently considered a pan-fungal barcode [33,34,35]. This region is also frequently used for the identification of fungi from the order Chaetothyriales, where the predominant agents of chromoblastomycosis and phaeohyphomycosis are found [1,2,36,37].

Furthermore, a recent study published by Fonseca in 2022 [30] highlighted the efficiency of the *ITS* region in a simple digestion RFLP assay for identifying species of dermatophytes of the genera *Trichophyton* sp. and *Microsporum* sp. A comparison between the results obtained through enzymatic restriction and morphological identification showed the greater precision of the molecular technique compared to macroscopic and microscopic analyses, emphasizing the importance of this technique in distinguishing cryptic species of filamentous fungi.

In this context, the molecular identification assays using ITS-RFLP conducted in this study have been proven to accurately identify the main species responsible for CBM and FEO, showing greater specificity when compared to morphological analyses methods. When we compare the RFLP results with those obtained by sequencing, we observe that simple digestion was able to accurately identify 95.8% of the species used in this study, while double digestion accurately identified 100% of the agents, making this technique promising for the molecular identification of these species without the need for sequencing.

Therefore, the use of multiple enzymes resulted in more precise outcomes, attributed to the increased fragmentation of the target region, providing the better differentiation of genetically close species such as *Fonsecaea monophora*, *Fonsecaea pedrosoi*, and *Cladophialophora bantiana*, all belonging to the same phylogenetic clade [2].

In this context, ITS-RFLP accurately identified two significant species responsible for chromoblastomycosis: *F. pedrosoi* and *F. monophora*, both belonging to the genus *Fonsecaea* [2,38]. *Fonsecaea* sp. Is responsible for over 80% of the chromoblastomycosis cases worldwide [3]. Specifically, *Fonsecaea pedrosoi* is responsible for 84.1% of the registered cases in Latin America and the Caribbean [7].

Less prevalent chromoblastomycosis agents, such as *Cladophialophora carrioni*, *Fonsecaea pugnacius*, and *Fonsecaea nubica* [3,6,7], were exclusively assessed in silico through ITS-RFLP, as depicted in Appendix A. Notably, these agents displayed a distinct fragmentation pattern compared to the species examined in vitro in this study. Within this context, it is imperative to acknowledge the potential limitations of the proposed technique in identifying these less incident species, thus warranting further investigations for the refinement and expansion of this methodology.

Besides the agents of chromoblastomycosis, it is possible to identify two species responsible for phaeohyphomycosis. The first is *Exophiala dermatitidis*, a causative agent of cutaneous, subcutaneous, and invasive infections [26,39,40], which, in severe cases, can affect the central nervous system [41]. The second identifiable species is *Cladophialophora bantiana*, a causative agent of PHM with high lethality due to invasive infections, mainly affecting the central nervous system of immunocompromised and post-transplant patients [25,40,41,42].

It is noteworthy that phaeohyphomycosis is caused by a variety of dematiaceous fungi with extensive genetic variation [1], posing challenges in identifying numerous species in a single RFLP assay. This limitation underscores the significance of phenotypic identification methods and emphasizes the necessity for further studies contributing to the development of novel, cost-effective molecular techniques for the identification of these fungal pathogens.

## 5. Conclusions

Our results introduce a novel molecular identification method based on ITS-RFLP for the two major causative agents of chromoblastomycosis worldwide and for two significant agents causing opportunistic and invasive phaeohyphomycosis.

As the causative agents of these infections differ in terms of virulence, pathogenicity, tropism, and susceptibility to antifungals, necessitating their identification for a more accurate prognosis of the infection and guide appropriate therapy, preventing serious complications such as infection dissemination or an inadequate treatment response.

In this context, the identification of these species through ITS-RFLP expands the range of molecular identification possibilities for these agents, providing rapid results at a lower cost compared to sequencing, only necessitating a laboratory infrastructure capable of performing basic molecular biology techniques such as PCR. This enables the improvement of accuracy and speed in diagnosis, ensuring the proper management of these fungal infections.

## Figures and Tables

**Figure 1 jof-10-00159-f001:**
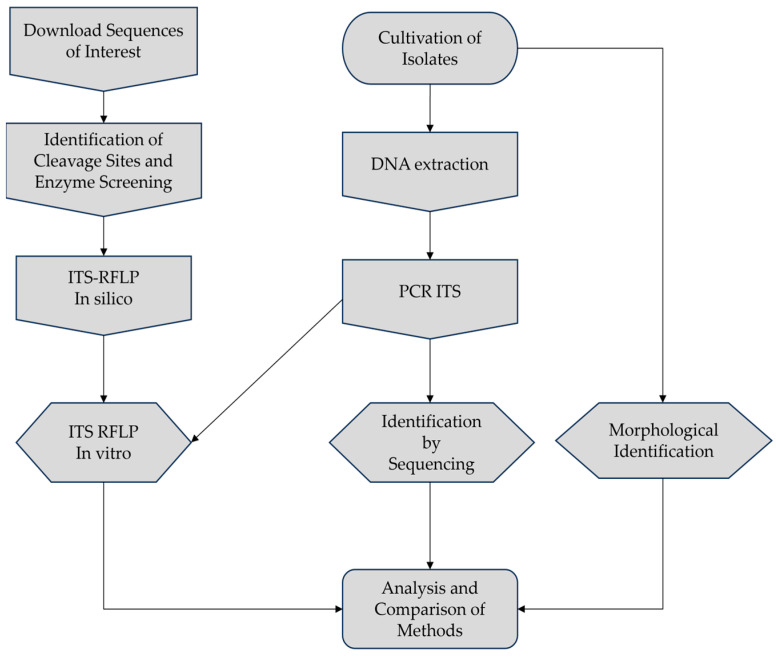
Workflow employed in this study.

**Figure 2 jof-10-00159-f002:**
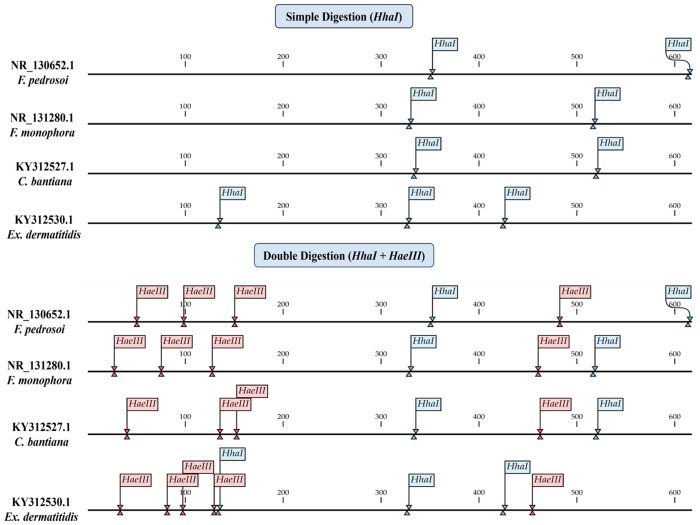
In silico analysis of the cleavage site positions for the Hhal and HaeIII enzymes, identifying the locations of the sites and the organization of fragmentation in single and double digestion.

**Figure 3 jof-10-00159-f003:**
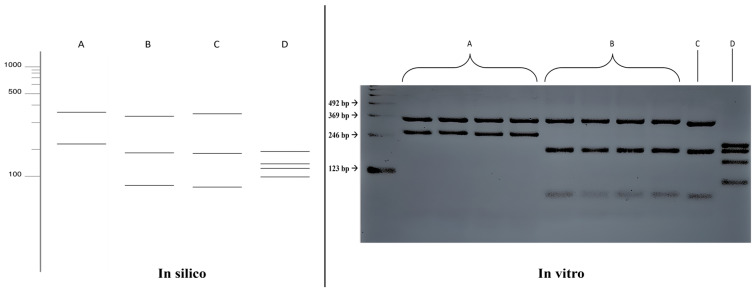
Comparison of fragmentation patterns obtained with enzymatic restriction using HhaI in silico and in vitro. A: *Fonsecaea pedrosoi*; B: *Fonsecaea monophora*; C: *Cladophialophora bantiana*; D: *Exophiala dermatitidis*.

**Figure 4 jof-10-00159-f004:**
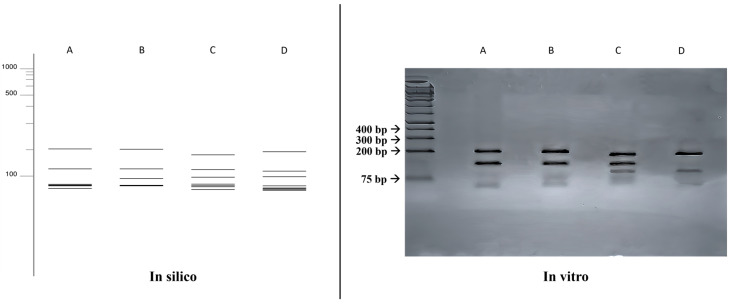
Comparison of fragmentation patterns obtained with enzymatic restriction using HhaI and HaeIII. A: *Fonsecaea monophora*; B: *Fonsecaea pedrosoi*; C: *Cladophialophora bantiana*; D: *Exophiala dermatitidis*.

**Table 1 jof-10-00159-t001:** Isolates causing chromoblastomycosis and phaeohyphomycosis used in this study.

Isolate	Genbank	Species	Anatomical Site	Host	Geographic Origin
IEC-CBM02	KY312523	*F. pedrosoi*	Thigh	Human	Pará/Brazil
IEC-CBM03	KY312524	*F. pedrosoi*	Foot	Human	Pará/Brazil
IEC-CBM04	KY312525	*F. pedrosoi*	Leg	Human	Pará/Brazil
IEC-CBM05	KY312526	*F. pedrosoi*	Leg	Human	Pará/Brazil
IEC-CBM06	KY312527	*C. bantiana*	CNS †	Human	Pará/Brazil
IEC-CBM07	KY312528	*F. pedrosoi*	Hand	Human	Pará/Brazil
IEC-CBM08	KY312529	*F. pedrosoi*	Leg	Human	Pará/Brazil
IEC-CBM09	KY312530	*E. dermatitidis*	*	Human	Pará/Brazil
IEC-CBM10	KY312531	*F. pedrosoi*	Arm	Human	Pará/Brazil
IEC-CBM11	KY312532	*F. pedrosoi*	Leg	Human	Pará/Brazil
IEC-CBM12	KY312533	*F. pedrosoi*	Foot	Human	Pará/Brazil
IEC-CBM13	KY312534	*F. monophora*	Leg	Human	Pará/Brazil
IEC-CBM14	KY312535	*F. pedrosoi*	*	Human	Pará/Brazil
IEC-CBM15	KY312536	*F. pedrosoi*	*	Human	Pará/Brazil
IEC-CBM16	KY312537	*F. pedrosoi*	Thigh	Human	Pará/Brazil
IEC-CBM17	KY312538	*F. monophora*	Leg	Human	Pará/Brazil
IEC-CBM18	MF416919	*F. pedrosoi*	Thigh	Human	Pará/Brazil
IEC-CBM19	MF416920	*F. monophora*	Forearm	Human	Pará/Brazil
IEC-CBM21	MF416922	*F. pedrosoi*	Fist	Human	Pará/Brazil
IEC-CBM22	MF416923	*F. pedrosoi*	Ankle	Human	Pará/Brazil
IEC-CBM23	MF416924	*F. monophora*	*	Human	Pará/Brazil

* Denotes unknown information; † central nervous system.

**Table 2 jof-10-00159-t002:** Comparison of identification results between morphological analysis, ITS-RFLP (HhaI), ITS-RFLP (HhaI + HaeIII), and sequencing of the ITS region.

Sample	Morphological Identification	ITS-RFLP (HhaI)	ITS-RFLP (HhaI + HaeIII)	Sequencing Identification
IEC-CBM02	*Fonsecaea* sp.	*F. pedrosoi*	*F. pedrosoi*	*F. pedrosoi*
IEC-CBM03	*Fonsecaea* sp.	*F. pedrosoi*	*F. pedrosoi*	*F. pedrosoi*
IEC-CBM04	*Fonsecaea* sp.	*F. pedrosoi*	*F. pedrosoi*	*F. pedrosoi*
IEC-CBM05	*Fonsecaea* sp.	*F. pedrosoi*	*F. pedrosoi*	*F. pedrosoi*
IEC-CBM06	*Cladophialophora* sp.	*C. bantiana/* *F. monophora* *	*C. bantiana*	*C. bantiana*
IEC-CBM07	*Fonsecaea* sp.	*F. pedrosoi*	*F. pedrosoi*	*F. pedrosoi*
IEC-CBM08	*Fonsecaea* sp.	*F. pedrosoi*	*F. pedrosoi*	*F. pedrosoi*
IEC-CBM09	*Exophiala* sp.	*E. dermatitidis*	*E. dermatitidis*	*E. dermatitidis*
IEC-CBM10	*Fonsecaea* sp.	*F. pedrosoi*	*F. pedrosoi*	*F. pedrosoi*
IEC-CBM11	*Fonsecaea* sp.	*F. pedrosoi*	*F. pedrosoi*	*F. pedrosoi*
IEC-CBM12	*Fonsecaea* sp.	*F. pedrosoi*	*F. pedrosoi*	*F. pedrosoi*
IEC-CBM13	*Fonsecaea* sp.	*F. monophora*	*F. monophora*	*F. monophora*
IEC-CBM14	*Fonsecaea* sp.	*F. pedrosoi*	*F. pedrosoi*	*F. pedrosoi*
IEC-CBM15	*Fonsecaea* sp.	*F. pedrosoi*	*F. pedrosoi*	*F. pedrosoi*
IEC-CBM16	*Fonsecaea* sp.	*F. pedrosoi*	*F. pedrosoi*	*F. pedrosoi*
IEC-CBM17	*Fonsecaea* sp.	*F. monophora*	*F. monophora*	*F. monophora*
IEC-CBM18	*Fonsecaea* sp.	*F. pedrosoi*	*F. pedrosoi*	*F. pedrosoi*
IEC-CBM19	*Fonsecaea* sp.	*F. monophora*	*F. monophora*	*F. monophora*
IEC-CBM21	*Fonsecaea* sp.	*F. pedrosoi*	*F. pedrosoi*	*F. pedrosoi*
IEC-CBM22	*Fonsecaea* sp.	*F. pedrosoi*	*F. pedrosoi*	*F. pedrosoi*
IEC-CBM23	*Fonsecaea* sp.	*F. monophora*	*F. monophora*	*F. monophora*

* Inconclusive result due to similar patterns in two different species.

## Data Availability

Data are contained within the article and Appendix A.

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
