# Peer review of "Identification of Chromoblastomycosis and Phaeohyphomycosis Agents through ITS-RFLP"

_jof, 2024, doi:10.3390/jof10020159_

Round 1
Reviewer 1 Report
Comments and Suggestions for Authors
The study focuses on the challenges associated with the morphological similarity of CBM and FEO agents, leading to the necessity of molecular techniques for accurate identification. The authors propose a methodology based on PCR-RFLP using the ITS region for identifying potential restriction enzyme sites. The study includes in silico analyses, DNA extraction, morphological identification, sequencing, and in vitro validation of the PCR-RFLP method. The results demonstrate the efficacy of the proposed assay in accurately identifying major CBM and FEO agents, providing a promising tool for diagnosis and proper management of these infections.
Overall: CBM and FEO are significant fungal infections, and the development of a reliable molecular identification method addresses a crucial need in the field of medical mycology.
Minor comments
- The article lacks a discussion on the limitations of the proposed method. Addressing potential challenges or limitations would strengthen the study's transparency and applicability.
- Figures illustrating the experimental setup, workflow, or graphical representations of results could enhance the visual appeal and clarity of the article.
Author Response
For research article
|
Response to Reviewer X Comments
|
||
|
1. Summary |
|
|
|
Thank you very much for taking the time to review this manuscript. Please find the detailed responses below and the corresponding revisions/corrections highlighted/in track changes in the re-submitted files. |
||
|
2. Questions for General Evaluation |
Reviewer’s Evaluation |
Response and Revisions |
|
Does the introduction provide sufficient background and include all relevant references? |
Not applicable |
|
|
Are all the cited references relevant to the research? |
Yes |
|
|
Is the research design appropriate? |
Yes |
|
|
Are the methods adequately described? |
Yes |
|
|
Are the results clearly presented? |
Yes |
|
|
Are the conclusions supported by the results? |
Yes |
|
|
3. Point-by-point response to Comments and Suggestions for Authors |
||
|
Comments 1: The article lacks a discussion on the limitations of the proposed method. Addressing potential challenges or limitations would strengthen the study's transparency and applicability. |
||
|
Response 1: Thank you for pointing this out. We agree with this comment. Therefore, we have added two new paragraphs in the discussion to elucidate the limitations of the proposed method. In the first paragraph, located on page 10 and starting at line 272, we address the limitations and challenges of the method for identifying chromoblastomycosis agents. Currently, our method is restricted to the two most prevalent species, as we do not have isolates of other chromoblastomycosis agents in our culture collection. Additionally, obtaining isolates of interest for the study from other institutions proved challenging. Consequently, the identification of less prevalent chromoblastomycosis agents was confined to in silico studies, as illustrated in Supplementary Figure 1, which we are submitting herewith. The second added paragraph, located on page 10 and commencing at line 286, further emphasizes the limitation of the proposed technique in identifying other species causing phaeohyphomycosis due to the extensive variety of fungi responsible for this infection. This diversity complicates the development of an RFLP-based method for many phaeohyphomycosis agents, given the substantial genetic variability among them. The paragraph underscores the importance of complementary identification methods, such as phenotypic approaches, to accurately determine other species. Additionally, it emphasizes the need for expanded studies contributing to the implementation of novel molecular techniques. Lastly, a brief explanation has been added on page 10 in the third paragraph of the conclusion, starting at line 302, highlighting the necessary structural requirements for the application of the proposed method.
“Less prevalent chromoblastomycosis agents, such as Cladophialophora carrioni, Fonsecaea pugnacius, and Fonsecaea nubica, were exclusively assessed in silico through ITS-RFLP [3, 6-7] as depicted in Figure S1. Notably, these agents displayed a distinct fragmentation pattern compared to the species examined in vitro in this study. Within this context, it is imperative to acknowledge potential limitations of the proposed technique in identifying these less incident species, thus warranting further investigation for the refinement and expansion of this methodology.”
“It is noteworthy that phaeohyphomycosis is caused by a variety of dematiaceous fungi with extensive genetic variation [1], posing challenges in identifying numerous species in a single RFLP assay. This limitation underscores the significance of phenotypic identification methods and emphasizes the necessity for further studies contributing to the development of novel, cost-effective molecular techniques for the identification of these fungal pathogens” |
||
|
Comments 2: Figures illustrating the experimental setup, workflow, or graphical representations of results could enhance the visual appeal and clarity of the article. |
||
|
Response 2: We appreciate the suggestion and agree with it. In this context, to enhance the clarity and visual appeal of the article, we have created a workflow diagram illustrated in Figure 1 on page 2. Additionally, we replaced Table 2 with Figure 2, now positioned on page 6. In this figure, we have visually represented the cleavage sites along the studied sequences, both in single and double digestion, for a more intuitive understanding. |
||
Reviewer 2 Report
Comments and Suggestions for Authors
It is a good paper, since it allows us to identify the causative agents of chromoblastomycosis and invasive pheohyphomycosis using the ITS-RFLP method.
The main question addressed by the research is to Identify agents of chromoblastomycosis and pheohyphomycosis, using molecular biology techniques suitable for laboratories with limited resources. CBM is highly prevalent and PHM is reported in various nations. Currently, these mycoses are difficult to diagnose due to their phenotypic similarities and their analyzes are subjective to the operator. The present study develops a molecular technique based on the restriction of fragment length polymorphism of the ITS region for the identification of species among the main causative agents of chromoblastomycosis and pheohyphomycosis.
The authors should increase the number of CBM and PHM isolates examined in the study. On the other hand, they must specify the sequences they use according to the agents studied.
Author Response
For research article
|
Response to Reviewer X Comments
|
||
|
1. Summary |
|
|
|
Thank you very much for taking the time to review this manuscript. Please find the detailed responses below and the corresponding revisions/corrections highlighted/in track changes in the re-submitted files. |
||
|
2. Questions for General Evaluation |
Reviewer’s Evaluation |
Response and Revisions |
|
Does the introduction provide sufficient background and include all relevant references? |
Yes |
|
|
Are all the cited references relevant to the research? |
Yes |
|
|
Is the research design appropriate? |
Yes |
|
|
Are the methods adequately described? |
Yes |
|
|
Are the results clearly presented? |
Yes |
|
|
Are the conclusions supported by the results? |
Yes |
|
|
3. Point-by-point response to Comments and Suggestions for Authors |
||
|
Comments 1: The authors should increase the number of CBM and PHM isolates examined in the study |
||
|
Response 1: Thank you for pointing this out. Unfortunately, we do not have additional isolates in our culture collection, and obtaining isolates from other institutions has proven challenging. We attempted to alleviate this issue by incorporating a substantial number of sequences in the in silico analyses. Our goal was to broaden our in vitro testing samples with additional isolates from other species as well. However, due to the aforementioned issues and the challenges in acquiring new clinical isolates because of the epidemiological aspects inherent in these infections, we were unable to do so. |
||
|
Comments 2: On the other hand, they must specify the sequences they use according to the agents studied. |
||
|
Response 2: Thank you for the suggestion. We agree with your recommendation and have added species identification to Table 1, located on page 3. Additionally, we have provided a supplementary table with GenBank codes, and the species used in the in silico analyses. This table is referenced on page 5, line 136. |
||